# Autoclave reactor synthesis of upconversion nanoparticles, unreported variables, and safety considerations
Rebecca McGonigle[1], Jodie Glasgow[1,2], Catriona Houston[1], Iain Cameron [1,2], Christian Homann [3], Dominic J. Black[4], Robert Pal [4] & Lewis E. MacKenzie [1] ✉

Autoclave reactors are widely used across chemical and biological sciences, including for the synthesis of upconversion nanoparticles (UCNPs) and other nanomaterials. Yet, the details of how autoclave reactors are used in such synthesis are rarely reported in the literature, leaving several key synthesis variables widely unreported and thereby hampering experimental reproducibility. In this perspective, we discuss the safety considerations of autoclave reactors and note that autoclaves should only be used if they are (a) purchased from reputable suppliers/manufacturers and (b) have been certified compliant with relevant safety standards. Ultimately, using unsuitable autoclave equipment can pose a severe physical hazard and may breach legal safety requirements. In addition, we highlight several parameters in autoclave synthesis that should be reported as standard to maximise the reproducibility of autoclave synthesis experiments across materials and chemistry research. We encourage users of autoclave synthesis vessels to: (1) adopt high-safety autoclaves and (2) report the many experimental variables involved to enhance experimental reproducibility.

In the context of chemical synthesis, autoclave reactors (sometimes simply referred to as "autoclaves") are extremely strong enclosed metal vessels designed to contain reaction mixtures at elevated temperatures and pressures. Autoclave reactors serve as the "pot" in which many hydrothermal and solvothermal reactions occur. Autoclave reactors have the advantage that they can be heated with common laboratory equipment such as hotplates or ovens. There are a variety of autoclave reactors available on the market, which can accommodate reactions of different scales and single or multiple reaction vessels. Autoclave reactors are used in many aspects of chemistry and materials science, including the synthesis of many types of nano- and micromaterials[1,2], metal-organic frameworks[1], catalysis[3], hydrogenation[4], polymerisation[5], materials testing[6], digestion[7], single-crystal casting[8], and corrosion testing[9]. Autoclave reactors have been widely used as a method of synthesis for upconversion nanoparticles (UCNPs) and are, therefore, the focus of this perspective.

UCNPs are inorganic crystalline nanostructures consisting of a low-photon energy host lattice doped with photonically active trivalent lanthanide ions. There are many possible host lattices for UCNPs[10–13], with $NaYF_4$ being the most common[14]. The long-lived excited states of the lanthanide ions (typically ~100 μs to ~10 ms)[15] enable multi-photon absorption and subsequent upconversion process, where multiple low-energy photons are absorbed and converted into a higher energy photon[16]. UCNPs can absorb and emit light at various wavelengths, primarily dependent on the choice of photoactive dopants hosted within the UCNP lattice structure. Sensitisers include $Yb^{3+}$ (~976 nm), $Nd^{3+}$ (~808 nm), and $Er^{3+}$ (~1532 nm)[17,18]. $Er^{3+}$ and $Tm^{3+}$ are common emissive activator ions, enabling emission from UV to visible at discrete "line-like" wavebands[19–21]. UCNP properties can be further enhanced via incorporating core/shell architectures. Carefully designed and synthesised photonically active core/shell architecture can enable advanced multi-wavelength excitation, e.g., for display technologies, and enhance quantum yield[18,22,23]. Core/shell architectures (e.g., layers of silica or the host lattice material) can also shield photoactive ions from solvent quenching that otherwise reduce UCNP emission[24]. Typical quantum yields for UCNPs in non-polar solvents are ~0.01% to 0.1%[25,26]. However, the greatest UCNP quantum yield to-date was 10.3% for core/shell hexagonal-phase $NaYF_4$:18%$Yb^{3+}$,2%$Er^{3+}$ for UCNPs in a dry form reported by Homann et al.[27]. For comparison, the quantum yield of rhodamine 6 G is 95% in EtOH; whereas 25–75% is typical for quantum dots[28,29]. Nevertheless, there are reports of bright luminescence from well-optimised UCNPs being visible to the naked eye[30,31]. UCNPs offer

[1]Department of Pure and Applied Chemistry, University of Strathclyde, Glasgow, G1 1RD, UK. [2]Department of Chemistry, University of Manchester, Manchester, M13 9PL, UK. [3]Division of Biophotonics, Federal Institute for Materials Research and Testing (BAM), 12489 Berlin, Germany. [4]Department of Chemistry, Durham University, Durham, DH1 3LE, UK. ✉e-mail: l.mackenzie@strath.ac.uk

several key advantages over other optically active materials, such as fluorescent dyes and quantum dots. Prominently, UCNPs do not photodegrade, photobleach, or blink[32–36]. This makes UCNPs well suited to applications such as nanoscale temperature sensing[37], pressure sensing[38,39], transparent multicolour volumetric display technologies[18,40], as well as nanopatterned security inks[41–43]. In biological environments, the diffuse NIR UCNP excitation can travel through several millimetres of blood[17], and several centimetres of ex vivo tissue due to minimal absorption and scattering at NIR wave ranges[44–46]. Further, NIR excitation does not induce visible autofluorescence, and has low phototoxicity[47]. The combination of these optical properties makes UCNPs uniquely suited to all-optical reporting in life sciences applications. Beyond purely optical properties, dopants also enable multi-modal responses, for example, incorporation of $Gd^{3+}$ and $Dy^{3+}$ within or onto UCNPs induces a parametric response for magnetic resonance imaging (MRI) contrast[48] complimentary to x-ray computed tomography (CT)[49], and incorporation of isotopes such as Fluorine-18 within UCNPs enables positron electron tomography (PET) response[50].

## Using autoclave reactors
### Advantages and disadvantages of autoclave reactors in comparison to other UCNP synthesis approaches

Given the utility of UCNPs and the expense of purchasing UCNPs from commercial vendors (see Supplementary Section B for the cost of purchasing UCNPs), it is often necessary for researchers to synthesise UCNPs for their research projects. A variety of UCNP synthesis approaches have been developed, the major ones being microwave reactions, hot-injection reactions, polymer-assisted reactions, and autoclave reactor approaches. Each method has advantages and disadvantages when it comes to (a) equipment and expertise required, (b) range of materials that can be produced, and (c) scalability. For brevity, these methods are discussed in depth in supplementary material (Sections D-G). A comparative summary of equipment and skill requirements, costs, and capabilities for each UCNP synthesis method is provided in Table 1.

Use of an autoclave reactor enables hydrothermal and solvothermal synthesis of UCNPs at moderate costs in comparison to other approaches (see Table 1), and can give rise to a wide range of UCNP morphologies based upon various lattice materials (see Supplementary Figs. S4 and S5 for examples). In such reactions, the autoclave reactor acts to safely contain any pressure build-up arising from the gas pressure of the reaction mixture at elevated temperatures (typically 180 °C for standard autoclave reactors). For contextual information regarding the hydrothermal and solvothermal synthesis of oleic-acid capped UCNPs (OA-UCNPs) and polyethyleneimine capped UCNPs (PEI-UCNPs), the reader is referred to the case studies provided in Supplementary Sections H and I. However, there are some disadvantages to autoclave reactor synthesis. For example, control of the heating rate and cooling rate is generally limited, and many autoclave reactors do not accommodate internal temperature monitoring. Further, scaling up autoclave reactions requires adopting either a parallel synthesis approach (e.g., multiple small-capacity autoclave reactors) or a much larger autoclave reactor; both approaches involve considerable expense.

Given the temperature and pressure hazards, autoclave reactor synthesis requires (1) careful attention to safety (see sections "Autoclave design, safety, and operation"), (2) an appreciation of the related legal regulations (section "Legal considerations of autoclaves"), and (3) an understanding of the experimental variables which are often unreported in the literature (section "The unreported variables in autoclave synthesis").

### Autoclave design, safety, and operation

Safety is a major concern with pressurised autoclave reactors because an autoclave failure or erroneous opening under pressure can result in the release of large amounts of energy (i.e., an explosion). The hazards associated with autoclave reactor synthesis are compounded by: (a) the design of the autoclave reactor and (b) the type of reaction occurring and the resultant pressure build-up. In hydrothermal UCNP synthesis reactions approaching 100 °C, water will, of course, form steam, which generates an elevated

pressure (we typically record 10 to 25 bars of pressure in such hydrothermal syntheses). In solvothermal reactions, UCNP synthesis will only generate excess pressure if heated to temperatures in excess of the boiling point of the solvents used, e.g., ethylene glycol [197 °C], oleic-acid [286 °C], and 1-octadecene (ODE) [178–179 °C][51–53]. Therefore at typical autoclave reactor synthesis temperatures (i.e., ~180–200 °C), solvothermal synthesis (assuming dry starting products) will not generate significant elevated pressures.

Autoclave reactor designs range from simple screw-thread systems to more sophisticated high-safety systems incorporating multiple redundant safety features. In general, autoclave reactors will include a liner (e.g., PTFE or suitable glass), to hold the reaction mixture. The simplest varieties of autoclaves may feature an over-pressure disk. However, if the autoclave reactor does not have a pressure gauge or internal temperature reading system, then one has to rely on the good judgement of the operator. Additionally, care must be taken with screw-thread systems to preserve the integrity of screw threads and to ensure there is no excess pressure within the autoclave reactor before it is opened. Therefore, simple autoclave reactors have higher risks of accidental failure and more advanced high-safety autoclave reactors are preferable.

Advanced high-safety autoclave reactor systems will feature engineering controls to help ensure safety. As a minimal example, the Berghoff DAB pressure vessels (see Fig. 1a) include a rupture disk to release pressure build-ups beyond design tolerances[54]. Both the Berghoff DB series (not shown) and Asynt PressureSyn series (see Fig. 1b) offer high-safety reactors that use clamps instead of threads, pressure release valves, and over-pressure bursting disks, temperature probe ports, and pressure gauges[55,56]. These features allow operators to identify and control potential pressure hazards during operation. Therefore, there is a lower risk of misuse.

Autoclaves are typically single-chamber apparatus, but it can be desirable to scale up synthesis via parallel batch production or simply larger autoclaves (e.g., autoclaves as large as 500 L are currently commercially available from manufacturers such as Büchiglasuster). Some manufacturers provide inserts to turn a single-chamber autoclave into a multi-chamber system, whilst others offer purpose-designed multi-reactor systems. Such multi-chamber reactors can speed up iterative synthesis.

Autoclave reactors can be heated by numerous means, including simple ovens, heating blocks, and advanced heating systems. Indeed, advanced heating systems may provide advantages with regard to space efficiency, temperature control, and sensing of the reaction parameters. The most advanced autoclave reactor systems also enable automated temperature data logging and feature active cooling to recover products faster than passive cooling will allow, thereby increasing potential synthesis rate, and enabling safety features such as automatic shut-down if nearing maximum limits. Advanced systems may also enable magnetic stirring, which typically cannot be accommodated in an oven. Notably, the most advanced systems available remove the need for manual handling, thereby preventing burn hazards and allowing maximum accessibility to all users—after all, autoclaves are made from solid metal (e.g., stainless steel) and are, therefore, cumbersome to handle. These features can help ensure safety and to enable reproducible synthesis.

To summarise: contemporary high-safety autoclave reactor systems offer benefits in terms of (1) safety, (2) accessibility, and (3) reaction monitoring. However, they are more expensive than (arguably unsafe) simple screw-thread autoclave reactors. Further, more sophisticated autoclave reactor systems with larger or multiple reaction chambers may also help reduce the time required to iterate synthesis towards optimisation or simply enable the generation of a greater quantity of desired product.

Some additional "last line of defence" measures above standard laboratory procedures may be considered for best safety practices when using autoclave reactors. (1) A form of secondary shielding around the autoclave reactor system (e.g., polycarbonate shields) to protect partially against accidental discharge or over-pressure. (2) Provision of hearing protection for operators in case of rupture of an over-pressure safety release valve.

**Table 1 | Comparative summary of equipment needed for each UCNP synthesis method**

| Synthesis approach | Equipment needed | Typical peak reaction temperature | Approximate cost of required apparatus (new, 2024)[a] | Practical skills required | Amount of UCNPs produced | Ease of automation |
|---|---|---|---|---|---|---|
| PVP-assisted synthesis | Hotplate/stirrer, beakers, ventilation. | 160 °C typical. | ~£1000 | Standard chemistry laboratory competence. | ~130 mg per reaction. | Batch production is possible via multiple flasks. |
| Microwave synthesis | Hotplate/stirrer and beakers to prepare precursors, laboratory grade microwave reactor, ventilation. | 260 °C | ~£20,000+ | Standard chemistry laboratory competence. Familiarity with microwave reactor. | ~200 mg per 18 mL reaction vessel. | Batch production via multi-reactor systems. |
| Autoclave synthesis | Hotplate/stirrer and beakers to prepare precursors, high-safety autoclave reactor, ventilation. | 180–200 °C for typical autoclave apparatus. | ~£12,000+ | Standard chemistry laboratory competence. Familiarity with autoclave reactor. | ~130 mg per reaction. | Parallel batch production via multi-reactor systems. |
| Hot-injection "thermal decomposition" "thermal co-precipitation" | Heating mantle, Schlenk line and vacuum systems, syringe pumps, ventilation. | 360 °C | ~£6,000 for typical apparatus. | Schlenk line operation. Inert atmosphere chemistry . | Up to 60 g per batch demonstrated[67]. | Automatic laboratory systems[19,20]. |

[a]Pricing from reputable commercial suppliers in the UK in 2024. Not including costs for ventilation or servicing.

The maximum heat and pressure a given autoclave reactor can safely sustain is primarily determined by the materials it is manufactured from; this is often stainless steel, with other options available on the market including alloys, glass, nickel, titanium, and zirconium options available for various temperature ranges, including temperature in excess of >200 °C. However, it is worth noting that PTFE liners are known to soften and deform if exposed to sufficiently high temperatures and pressurises arising from erroneous usage (see Fig. 1c). Instead, liners made of borosilicate glass may be better suited to high-temperature reactions.

### Legal considerations of autoclaves

There are various regulations worldwide regarding the use of pressurised equipment. In the UK, employers have a legal duty to comply with the "Pressure Systems Safety Regulations 2000 (PSSR)" as part of the broader legal duties specified in the "Health and Safety at Work Act" (1974)[57–59]. The PSSR regulation aims *"to prevent serious injury from the hazard of stored energy as a result of the failure of a pressure system or one of it's component parts"*, and applies to any *"compressed or liquefied gas, including air, at a pressure than greater than 0.5 above atmospheric pressure"* and *"pressurised hot water above 110 °C"* [57]. There are some exceptions to the PSSR, which include (but are not limited to) pressure systems to be used for "weapons systems", "vehicle tyres", and "experimental research". However, we note that most research is conducted at universities that have a duty of care to their students and staff, so we suggest that it would be good practice to abide by the PSSR when using autoclaves for nanomaterial synthesis research. In the European Union, the directive 2014/68/EU[60] governs the certification and testing of equipment pressurised to >0.5 bar[60]. Guidance in other countries varies and cannot be comprehensively covered here. We encourage all users of autoclave reactors to (a) familiarise themselves with the legal requirements and guidance regarding autoclave reactors for their specific country and (b) to source autoclaves that are compliant with the highest international standards.

Autoclave reactors should only ever be purchased from reputable scientific suppliers who pre-test and certify their autoclave reactors to governmental standards. Examples of reputable manufacturers include (but may not be limited to): Berghof GmBH (Germany), Büchiglasuster (Switzerland), Asynt Ltd (United Kingdom), LBBC Baskerville Ltd (United Kingdom), Mettler Toledo (USA/global), and Parr Instrument Company (USA). It is also worth noting that, in our experience, some manufacturers will provide certification of testing but may not provide a user manual or equivalent example operating procedures. This can result in some unexpected issues for inexperienced users. For example, any ferrules that are used to introduce thermocouple probes to autoclave reactor ports will likely become permanently conjoined with a thermocouple probe after autoclave usage. Further, appropriate high-temperature, high-pressure grease (e.g., CRC Lithium Grease 30570 for temporary operation up to 200 °C) will aid smooth situation of clamp components, and non-flammable leak detector fluid (e.g., SNOOP®) can be beneficial to check that all fittings are secure and pressure is contained.

It should be noted that there are some dubious low-cost autoclave reactors readily available via non-reputable manufacturers. These autoclave reactors will likely not be compliant with government standards at these low-price points. This could introduce a high risk of spurious failure and potentially lethal hazards. Therefore such low-cost autoclaves should not be used. We strongly recommend that research teams purchase autoclaves from certified reputable suppliers and ensure that their autoclave equipment meets appropriate governmental certification requirements.

### The unreported variables in autoclave synthesis

Many studies report autoclave UCNP synthesis, yet most of these studies do not report which autoclaves were used and how they were operated. For example, much of the literature simply states that reaction mixtures were

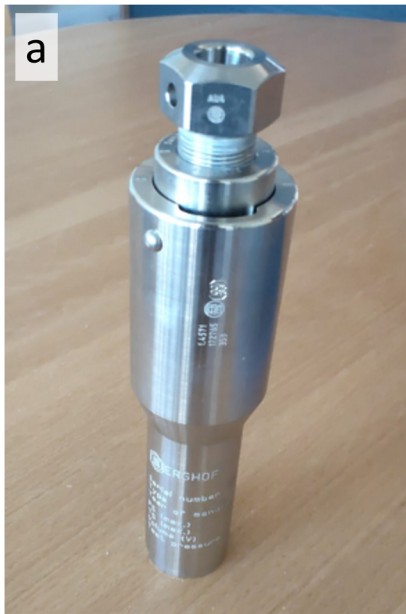
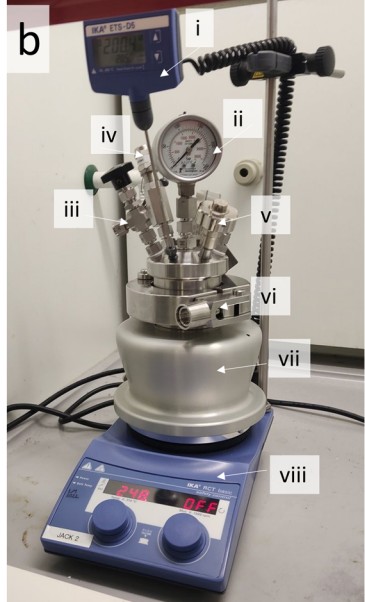
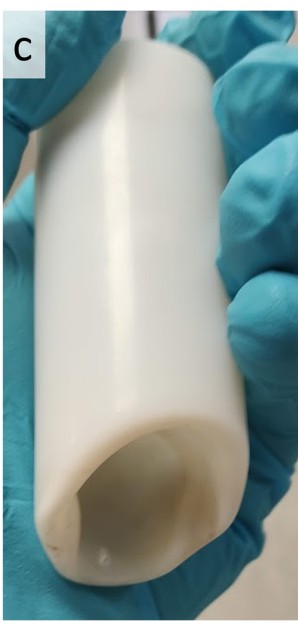

**Fig. 1 | Examples of autoclave reactor systems. a** A screw-thread Berghoff Digestec DAB autoclave reactor (photo courtesy of Dr Juliane Simmchen). **b** An assembled Asynt PressureSyn high-safety autoclave reactor and hotplate heating block situated on a hotplate stirrer within a fume hood. Labelled components: (i) temperature probe and heating controller. (ii) pressure gauge. (iii) pressure release valve. (iv) emergency over-pressure burst valve. (v) clamp key valve. (vi) clamp release mechanism which can only be released using the clamp key valve. (vii) heating block to ensure uniform heating. (viii) hotplate/stirrer for heating. **c** A PressureSyn PTFE liner, which was deformed due to the following process: (1) some liquid was erroneously left between the liner and metal autoclave, (2) this caused inefficient heating of the reaction mixture, resulting in the heating control system to apply a consistently elevated heating temperature, (3) resulting in excessive heating of the liquid, (4) elevating temperatures to a point where the PTFE softened and deformed, resulting in deformation of the bottom of the liner.

"*added to a Teflon-lined autoclave*" and "*heated at*" a temperature for some time. This raises many questions, such as:

- What volume is the autoclave reactor?
- How much internal space does the liner occupy?
- What is it made of?
- How is it heated?
- Was it pre-heated?
- Is the quoted temperature the internal reactor temperature or the externally applied heating temperature? These are often different—see Supplementary Fig. S3.
- What timepoint counts as reaction onset?
- Was the reaction stirred? What pressure did the reaction occur at?

This is a huge parameter space of unreported variables. In our experience, this combinatorial pitfall of variables can make it difficult for researchers to reproduce autoclave synthesis—particularly hydrothermal autoclave UCNP synthesis (see Supplementary Section H)—even when using the same autoclave equipment in the same laboratory environment. If it is challenging for researchers using the same equipment and reagents to reproduce, then how can we possibly reproduce UCNP synthesis studies where the equipment used is fundamentally unreported? The absence of comprehensive reporting has the potential to waste significant time and effort within our research community. Therefore, in the interest of robust, reproducible, and open science, we strongly recommend that a number of key autoclave parameters and variables be reported; these are summarised in Table 2.

For context, we have detailed two case studies of autoclave synthesis UCNPs. The first is hydrothermal synthesis of oleic-acid (OA) coated NaYF$_4$:Yb,Er,Mn UCNPs (976 nm excitation) (see Supplementary Section G). The second example is the synthesis of PEI-coated NaYF$_4$:Yb,Er@NaYF$_4$:Yb, Nd core/shell UCNPs featuring dual band 808 nm and 976 nm excitation in Supplementary Section H. These case studies are intended as illustrative examples of the importance of the aforementioned variables in autoclave synthesis, rather than comprehensive scientific reports.

### Future directions for autoclave reactor synthesis of UCNPs

In the longer term, there may also be scope to improve the wider standards of autoclave reactor synthesis of UCNPs. For instance, the field may consider the adoption of more transparent standardised operating protocols and/or standardised apparatus. This could also facilitate inter-group comparator experiments, where for example, a standardised output/reference material is synthesised to demonstrate verified high-quality UCNP synthesis and/or to compare measurements between research groups[13]. For example, in the wider field of nanomaterials research, standardised approaches have been developed and proposed for nano-material synthesis[61], physiochemical properties[61], optical properties (e.g., quantum yield)[62], toxicity[61,63,64], dynamic light scattering (hydrodynamic diameter)[65]. Whilst full discussion relating to standardised UCNP reference nanomaterials is beyond the scope of this paper, it should be noted that standardisation is a complex problem in the wider field of nanomaterials which requires input from the relevant global scientific communities[64,66].

### Outlook

A large number of publications have reported the use of autoclave reactors for the synthesis of nanomaterials, including UCNPs. However, to date, many important experimental and operational parameters associated with autoclave reactor equipment have not been reported. The lack of these crucial experimental details reduces experimental reproducibility across the nanomaterial literature. Herein, we have provided suggestions for experimental parameters that can and should be reported in detail of autoclave reactor synthesis procedures (see Table 2). We recommend that these details be provided wherever practicable in order to enhance future experimental reproducibility. We have also summarised some regulations and legalities covering the use of autoclaves for research and work purposes and have noted that these legalities should be carefully considered by all researchers involved in autoclave reactor usage. We recommend that research should only use autoclaves made by reputable manufacturers and which are certified/tested to comply with

**Table 2 | Key parameters and variables to report in autoclave reactor synthesis in order to maximise reproducibility**

| | Key parameter/variable | Importance |
|---|---|---|
| General autoclave reactor operation | Make and model of autoclave reactor. | Reproducibility and repeatability by others. |
| | Make and model of autoclave reactor heating system. | |
| | Maximum heat and pressure specified by manufacturers. | Understanding limits of reactor equipment. |
| | Reactor and sleeve cleaning procedure. | May help identify potential issues with contamination. |
| | Safety precautions. | Ensuring safety and sharing good practice. |
| Reaction containment and liner | Autoclave reactor chamber volume when empty. | Pressure of reaction depends on volume available for gas generation and expansion. |
| | Autoclave chamber volume once liner is inserted. | |
| | Volume of liner. | |
| | Total volume of reaction mixture added within liner (i.e., fill factor). | |
| | Liner material. | Influences heat transfer rate. |
| | Does the liner have a lid? | If not fully enclosed, gas may expand into autoclave headspace; possible contamination risk. |
| | Was an inert atmosphere introduced? | Inert atmospheres may benefit UCNP luminescence. |
| Stirring | Was the reaction mixture stirred? If so, at what rate? | May alter the size of any micelles present in hydrothermal reactions. |
| | What size and shape of stir bar was used? | Influences chamber volume. |
| Heating and temperature | Autoclave reactor construction materials. | Construction materials influence heat transfer rate. |
| | Heat source (e.g., oven, hotplate, heating mantle, etc). | Governs heating uniformity and heating rate. |
| | Heating control method and target temperature. | Heat delivery rate is a key variable in synthesis. |
| | Is the heating control and target temperature based on internal or external temperature? | External temperature and reactor internal temperature are different due to heat differentials. Reactor internal temperature lags behind external temperature. |
| | Was the reactor pre-heated? | Influences the heating rate and total energy in the system. |
| | When is the reaction timer deemed to have started? | Determines total energy input into the reaction mixture. |
| | Was temperature data logged as the reaction occurred? | Aids repeatability and troubleshooting. |
| | Heating rate. | Aids repeatability and troubleshooting. |
| Pressure | Does the autoclave reactor have a pressure gauge? | Pressure generated may be important for synthesis outcomes. |
| | What was the internal reactor pressure during the reaction? | Pressure generated may be important for synthesis outcomes. |

appropriate international regulatory standards. Further, we note that currently available high-safety autoclave reactor systems offer advantages in terms of safety, experimental control, and reproducibility.

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

## Acknowledgements

The picture of an autoclave used in Fig. 1a was kindly provided by our colleague Dr Juliane Simmchen (Department of Pure and Applied Chemistry, University of Strathclyde, Glasgow, UK). This research was supported by a Royal Society Research Grant (RGS\R1\221139), and a Royal Society of Chemistry Research Enablement Grant (E21-5833576777). L.M. was supported by a BBSRC Discovery Fellowship from 2020 to 2023 (BB/T009268/1). R.P. was supported by BBSRC grants (BB/S017615/1) and (BB/X001172/1).

## Author contributions

R. McGonigle: development of UCNP synthesis methods; creation of Fig. S2 and Fig. S5; edited and approved the manuscript. J. Glasgow: development of UCNP synthesis methods; creation of Fig. S3; edited and approved the manuscript. C. Houston: development of UCNP synthesis methods; edited and approved the manuscript. I. Cameron: testing of UCNP synthesis and functionalisation methods; edited and approved the manuscript. C. Homan: contextualisation of UCNP synthesis methods; key conclusions, drafted, edited, and approved the manuscript. D. Black: UCNP characterisation and testing; edited and approved the manuscript. R. Pal: UCNP characterisation and testing; edited and approved the manuscript. L.E. MacKenzie: sourced funding, conceived and drafted the manuscript; created all other figures, edited and proofed the manuscript.

## Competing interests

The authors declare no competing interests.
