## [Transparent Peer Review file · Communications Chemistry]

Autoclave reactor synthesis of upconversion nanoparticles, unreported variables, and safety considerations

Corresponding Author: Dr Lewis MacKenzie

Version 0:

Reviewer comments:

Reviewer #1

(Remarks to the Author)

This perspective on the autoclave synthesis of upconversion nanoparticles (UCNPs) is principally an interesting and timely idea. However, the concept of the paper does not really exploit the full potential of this idea. Generally, the paper is not well focussed. Another weakness presents the sometimes arbitrary information on out-of-scope topics, while other relevant information is missing. For example, in the introduction topics like strategies to increase of upconversion luminescence and brightness are mentioned, and this in a somehow arbitrary manner/referencing arbitrarily chosen publications although the direct link to the topic of the perspective is missing. In addition, the parameter excitation power density is not mentioned, which could also be utilized to enhance the upconversion luminescence. When referring to the properties of UCNPs, for example, the emission of several activator-characteristic luminescence bands, the spectral position of which is not affected by the environment is missing as well as their spectral shifting features including multiphotonic (and hence excitation power density dependent) upconverted and linear downshifted luminescence. When comparing UCNPs to other molecular or nanocrystalline luminophores, some numbers on typical molar absorption coefficients / absorption cross sections and quantum yields would have been informative.

In my opinion, the sections on historical trends and commercial UCNPs are not needed. Figure 3 is misleading and I find the idea of this scheme not really appealing. Here a list of the equipment required for the different syntheses which should have briefly introduced in this context would have been better and more scientifically solid.

Also some of the syntheses listed are not really needed and their presentation is questionable like sections 2.1.2 and 2.2, that contain too many arbitrary details out of the initially claimed scope of the perspective.

The main part of the perspective, section 2.3 is also too lengthy with too many details and redundancies. It would have been sufficient to describe the synthesis including the most important synthesis parameters controlling UCNP size, shape, and quality, address challenges, and provide some recommendations on suitable instrumentation, safety measures, and well working protocols as well as requirements regarding the information to be given in publications.

This perspective, which generally suffers from a lack of logical structure and consistency, could have been easily shortened to half of its current length without the loss of important information.

Reviewer #2

(Remarks to the Author)

This manuscript reports a review on the autoclave synthesis and unreported variables and safety considerations, specifically focusing on upconverting nanoparticles (UCNPs). This aspect is not reported in the literature though there are plenty of reports on using autoclave synthesis for different nanoparticles. The authors mainly discussed the reproducibility of the synthesis and safety concerns as autogenous pressure is developed inside the autoclave due to the presence of solvents and heating at high temperature. However, there are few queries need to be resolved before the manuscript can be considered.

1. It is true that literature reports on Autoclave synthesis may not provide all information/details. However, considering UCNPs, the luminescence features are not much influenced by morphology or size except for the intensity. Particularly with NaYF₄, the two phases may show difference in the green/red emission ratio. In fact, autoclave method is used in the synthesis of many nanoparticles. A stronger emphasis needs to be provided on Why do the authors consider mainly UCNPs.
2. The main theme of the review is on Autoclave synthesis of UCNPs. There are discussions on other methods like thermal co-precipitation and microwave towards the end of the manuscript. It should be discussed in the earlier before discussion on autoclave.
3. Also, it should be followed by the merits and demerits of this system and how autoclave methods are important and widely

used and then discuss the details of the autoclave methods.

4. In connection, the main theme/objective of the review is on the autoclave synthesis, however, I feel the discussion on photophysical aspects of lanthanides and upconversion are little lengthy. This can be written more concise.

5. Few relevant references are missing. Authors can check these works , where how ligands help in the synthesis of 2 crystalline phases of NaYF₄ (<http://dx.doi.org/10.1002/slct.201600699>). Also, how core/shell/shell strategy to improve the QY of UCNPs. This is the first work on this strategy using Autoclave method (Adv. Func. Mater., 2009, 19, 2924).

Reviewer #3

(Remarks to the Author)

The manuscript titled "Lessons Learned in Autoclave Synthesis of Upconversion Nanoparticles: Unreported Variables and Safety Considerations" by Rebecca McGonigle et al. requires significant revision. Initially, there was confusion regarding its classification as a review or research article, which was clarified later as a research article. The manuscript needs thorough reorganization, particularly in terms of figures and writing style. Unnecessary schematics and cartoons should be removed, and the focus should be shifted towards presenting the results concisely. Therefore, I recommend rejecting this article in its current form.

Version 1:

Reviewer comments:

Reviewer #1

(Remarks to the Author)

This perspective on the autoclave synthesis of upconversion nanoparticles (UCNPs) is interesting and a timely idea. Moreover, the MS has been considerably improved by eliminated non-relevant information for the topic addressed. However, the focus still needs to be more clearly stated in the introduction and then more stringently followed throughout the MS although this has been improved by the revision. The introduction should be nevertheless still shortened, providing (listing) the most relevant properties of UCNPs as requested in the previous review and then only topics relevant for the scope of the MS should be addressed, thereby avoiding arbitray information on out-of-scope topics.

I do not see the need to include section 2.5, this is not relevant for the scope of the MS. Although the other syntheses could be still shortened. Here it would be helpful in the introduction to include a clear statement of the scope "autoclave synthesis " and why this synthesis is relevant with respect to other approaches which could then be listed. For the out of scope syntheses, it is sufficient to describe the synthesis procedure in 1-2 sentences and cite reviews. Then present the topic, "autoclave synthesis", describe its advantages compared to toher methods and then as mostly done, give some recommendations on suitable instrumentation, safety measures, and well working protocols as well as requirements regarding the information to be given in publications.

Reviewer #2

(Remarks to the Author)

The authors have tried to address all the queries raised by the reviewers and satisfactorily improved the manuscript. The revised manuscript is suitable for consideration in this journal.

Overall response to reviewer's comments

We would like to thank all three peer-reviewers for their valuable comments. We have taken your feedback on board and have made major revisions to this manuscript in order to improve the focus and clarity of the paper. In particular, we have restructured the manuscript. We have removed the tangential topics and case studies and instead moved these to supplementary information for the most curious readers. We have also changed the structure to emphasise the most crucial section about important aspects of autoclave synthesis of upconversion nanoparticles (now Section 3).

Our responses to individual reviewer comments are set out below.

Reviewer #1 (Remarks to the Author):

R1 C1. This perspective on the autoclave synthesis of upconversion nanoparticles (UCNPs) is principally an interesting and timely idea. However, the concept of the paper does not really exploit the full potential of this idea. Generally, the paper is not well focussed. Another weakness presents the sometimes arbitrary information on out-of-scope topics, while other relevant information is missing. For example, in the introduction topics like strategies to increase of upconversion luminescence and brightness are mentioned, and this in a somehow arbitrary manner/referencing arbitrarily chosen publications although the direct link to the topic of the perspective is missing.

Response to R1 C1. We have removed a great deal of content which was not strictly pertinent to the core theme of the manuscript. This ancillary information is now supplementary material, and so can be accessed by readers, but will no longer distract from the manuscript.

R1 C2. In addition, the parameter excitation power density is not mentioned, which could also be utilized to enhance the upconversion luminescence.

Response to R1 C2. We have now mentioned excitation power density in the fourth paragraph of the introduction and referred the reader to the following citation: Jones, Callum MS, Anna Gakamsky, and Jose Marques-Hueso. "The upconversion quantum yield (UCQY): a review to standardize the measurement methodology, improve comparability, and define efficiency standards." *Science and Technology of Advanced Materials* 22.1 (2021): 810-848.

R1 C3. When referring to the properties of UCNPs, for example, the emission of several activator-characteristic luminescence bands, the spectral position of which is not affected by the environment is missing as well as their spectral shifting features including multiphotonic (and hence excitation power density dependent) upconverted and linear downshifted luminescence.

Response to R1 C3. We have added further particulars regarding activators in the second paragraph of the introduction and have noted that the spectral position of upconversion emission is independent of environment. We have also mentioned down-shifted luminescence. The reader is referred to power-dependence in paragraph 4 of the introduction.

R1 C4. When comparing UCNPs to other molecular or nanocrystalline luminophores, some numbers on typical molar absorption coefficients / absorption cross sections and quantum yields would have been informative.

Response to R1 C4. We have added this information for quantum dots and fluorescent dye rhodamine 6G in paragraph 4 in the introduction.

R1 C5. In my opinion, the sections on historical trends and commercial UCNPs are not needed.

Response to R1 C5. These sections have been deleted from the manuscript and are now in supplementary material which is not essential to the reader.

R1 C6. Figure 3 is misleading and I find the idea of this scheme not really appealing.

Response to R1 C6. This figure has been deleted entirely.

R1 C7. Here a list of the equipment required for the different syntheses which should have briefly introduced in this context would have been better and more scientifically solid.

Response to R1 C7. We agree. We have added this information in a new table, Table 1 in Section 2.1.

R1 C8. Also some of the syntheses listed are not really needed and their presentation is questionable like sections 2.1.2 and 2.2, that contain too many arbitrary details out of the initially claimed scope of the perspective. The main part of the perspective, section 2.3 is also too lengthy with too many details and redundancies. It would have been sufficient to describe the synthesis including the most important synthesis parameters controlling UCNP size, shape, and quality, address challenges, and provide some recommendations on suitable instrumentation, safety measures, and well working protocols as well as requirements regarding the information to be given in publications. This perspective, which generally suffers from a lack of logical structure and consistency, could have been easily shortened to half of its current length without the loss of important information.

Response to R1 C8. We have taken the reviewer's feedback on board and made major edits to the manuscript, including a restructure. The manuscript is now considerably shorter. We hope the reviewer finds these numerous edits favourable.

Reviewer #2 (Remarks to the Author):

This manuscript reports a review on the autoclave synthesis and unreported variables and safety considerations, specifically focusing on upconverting nanoparticles (UCNPs). This aspect is not reported in the literature though there are plenty of reports on using autoclave synthesis for different nanoparticles. The authors mainly discussed the reproducibility of the synthesis and safety concerns as autogenous pressure is developed inside the autoclave due to the presence of solvents and heating at high temperature. However, there are few queries need to be resolved before the manuscript can be considered.

R2 C1. It is true that literature reports on Autoclave synthesis may not provide all information/details. However, considering UCNPs, the luminescence features are not much influenced by morphology or size except for the intensity. Particularly with NaYF₄, the two phases may show difference in the green/red emission ratio. In fact, autoclave method is used in the synthesis of many nanoparticles. A stronger emphasis needs to be provided on Why do the authors consider mainly UCNPs.

Response to R2 C1. We have focused on UCNPs because autoclave synthesis of UCNPs is already a substantial field with a vast number of publications, with hundreds of relevant papers in the literature. Whilst we would like to include more information about different fields, regrettably, our manuscript was already rightly criticised for its length, therefore we cannot add in substantially more information about the myriad sub-fields of nanomaterial research which incorporate autoclave synthesis and apply this to many different chemistries, where each unreported variable may play a key role. However, we agree that our conclusions are generalizable across different fields. Therefore:

1. We have re-written our abstract to emphasize broad applicability.
2. We have mentioned various fields that use autoclave apparatus in Section 3. Particularly “Autoclaves are used in many aspects of chemistry and materials science, including synthesis of many types of nano and micromaterials,^{112,113} metal organic frameworks,¹¹² catalysis,¹¹⁴ hydrogenation,¹¹⁵ polymerisation,¹¹⁶ materials testing,¹¹⁷ digestion,¹¹⁸ single-crystal casting,¹¹⁹ and corrosion testing.^{120”}
3. Our re-structuring has made the section on autoclave apparatus and its usage (section 3.1) and legalities (section 3.2) more clear and independent of UCNPs.
4. The case studies on synthesis of UCNPs have been deleted from the manuscript and moved to the supplementary information.
5. We have tried to de-emphasise UCNPs in the conclusions and move towards to generalist conclusions.

We hope these points are satisfactory for the reviewer.

R2 C2. The main theme of the review is on Autoclave synthesis of UCNPs. There are discussions on other methods like thermal co-precipitation and microwave towards the end of the manuscript. It should be discussed in the earlier before discussion on autoclave.

Response to R2 C2. We agree. We have re-organised the structure of the manuscript so that these sections are earlier in the manuscript.

R2 C3. Also, it should be followed by the merits and demerits of this system and how autoclave methods are important and widely used and then discuss the details of the autoclave methods.

Response to R2 C3. We agree. We have revised Section 3 to include the merits and demerits of the autoclave system in the first paragraph of Section 3.

R2 C4. In connection, the main theme/objective of the review is on the autoclave synthesis, however, I feel the discussion on photophysical aspects of lanthanides and upconversion are little lengthy. This can be written more concise.

Response to R2 C2. We have tried to make these aspects more concise. However, we have found this comment hard to action for several reasons. Firstly, reviewer 1 wanted the addition of more information regarding photophysical properties of UCNPs. Secondly, it is quite difficult to reduce the length of these sections because the photophysical properties of UCNPs are dependent on the synthesis approach of UCNPs. Therefore to reduce the length of the photophysics in the introduction further, we would require more specific feedback from the reviewers with regards to which information should be cut-out,

R2 C5. Few relevant references are missing. Authors can check these works , where how

ligands help in the synthesis of 2 crystalline phases of NaYF₄ (<http://dx.doi.org/10.1002/slct.201600699>). Also, how core/shell/shell strategy to improve the QY of UCNPs. This is the first work on this strategy using Autoclave method (Adv. Func. Mater., 2009, 19, 2924).

Response to R2 C5. We have now added these references.

Reviewer #3 (Remarks to the Author):

R3 C1. The manuscript titled "Lessons Learned in Autoclave Synthesis of Upconversion Nanoparticles: Unreported Variables and Safety Considerations" by Rebecca McGonigle et al. requires significant revision. Initially, there was confusion regarding its classification as a review or research article, which was clarified later as a research article. The manuscript needs thorough reorganization, particularly in terms of figures and writing style. Unnecessary schematics and cartoons should be removed, and the focus should be shifted towards presenting the results concisely. Therefore, I recommend rejecting this article in its current form.

Response to R3 C1. We have thoroughly reorganized the manuscript, including deletion of the offending figures. Results have been more concisely presented and the focus shifted. Some ancillary information has not been completely expunged, but rather moved to supplementary information so as to enhance the flow and readability of the manuscript.

Response to Round 2 of reviewers comments

I was very pleased to see that this manuscript was provisionally accepted after this second round of peer review.

As suggested in the second round of comments, we have endeavored to further streamline the manuscript and generally improve readability. We have achieved this by moving several sections to supplementary information and adding a new table.

I would like to once again sincerely thank the two peer-reviewers for once again reviewing the manuscript. Your time and professional input is very much appreciated.

Sincerely,

Lewis MacKenzie